# SPECFORMER: SPECTRAL GRAPH NEURAL NETWORKS MEET TRANSFORMERS

**Deyu Bo**[1], **Chuan Shi**[1*], **Lele Wang**[2], **Renjie Liao**[2*]
Beijing University of Posts and Telecommunications[1],
University of British Columbia[2]
{bodeyu, shichuan}@bupt.edu.cn, {lelewang, rjliao}@ece.ubc.ca

## ABSTRACT

Spectral graph neural networks (GNNs) learn graph representations via spectral-domain graph convolutions. However, most existing spectral graph filters are scalar-to-scalar functions, i.e., mapping a single eigenvalue to a single filtered value, thus ignoring the global pattern of the spectrum. Furthermore, these filters are often constructed based on some fixed-order polynomials, which have limited expressiveness and flexibility. To tackle these issues, we introduce Specformer, which effectively encodes the set of all eigenvalues and performs self-attention in the spectral domain, leading to a learnable set-to-set spectral filter. We also design a decoder with learnable bases to enable non-local graph convolution. Importantly, Specformer is equivariant to permutation. By stacking multiple Specformer layers, one can build a powerful spectral GNN. On synthetic datasets, we show that our Specformer can better recover ground-truth spectral filters than other spectral GNNs. Extensive experiments of both node-level and graph-level tasks on real-world graph datasets show that our Specformer outperforms state-of-the-art GNNs and learns meaningful spectrum patterns. Code and data are available at `https://github.com/bdy9527/Specformer`.

## 1 INTRODUCTION

Graph neural networks (GNNs), firstly proposed in (Scarselli et al., 2008), become increasingly popular in the field of machine learning due to their empirical successes. Depending on how the graph signals (or features) are leveraged, GNNs can be roughly categorized into two classes, namely spatial GNNs and spectral GNNs. Spatial GNNs often adopt a message passing framework (Gilmer et al., 2017; Battaglia et al., 2018), which learns useful graph representations via propagating local information on graphs. Spectral GNNs (Bruna et al., 2013; Defferrard et al., 2016) instead perform graph convolutions via spectral filters (*i.e.*, filters applied to the spectrum of the graph Laplacian), which can learn to capture non-local dependencies in graph signals. Although spatial GNNs have achieved impressive performances in many domains, spectral GNNs are somewhat under-explored.

There are a few reasons why spectral GNNs have not been able to catch up. First, most existing spectral filters are essentially scalar-to-scalar functions. In particular, they take a single eigenvalue as input and apply the same filter to all eigenvalues. This filtering mechanism could ignore the rich information embedded in the spectrum, *i.e.*, the set of eigenvalues. For example, we know from the spectral graph theory that the algebraic multiplicity of the eigenvalue 0 tells us the number of connected components in the graph. However, such information can not be captured by scalar-to-scalar filters. Second, the spectral filters are often approximated via fixed-order (or truncated) orthonormal bases, *e.g.*, Chebyshev polynomials (Defferrard et al., 2016; He et al., 2022) and graph wavelets (Hammond et al., 2011; Xu et al., 2019), in order to avoid the costly spectral decomposition of the graph Laplacian. Although the orthonormality is a nice property, this truncated approximation is less expressive and may severely limit the graph representation learning.

Therefore, in order to improve spectral GNNs, it is natural to ask: *how can we build expressive spectral filters that can effectively leverage the spectrum of graph Laplacian*? To answer this question,

---

*Co-corresponding authors

we first note that eigenvalues of graph Laplacian represent the frequency, *i.e.*, total variation of the corresponding eigenvectors. The magnitudes of frequencies thus convey rich information. Moreover, the relative difference between two eigenvalues also reflects important frequency information, *e.g.*, the spectral gap. To capture both magnitudes of frequency and relative frequency, we propose a Transformer (Vaswani et al., 2017b) based set-to-set spectral filter, termed *Specformer*. Our Specformer first encodes the range of eigenvalues via positional embedding and then exploits the self-attention mechanism to learn relative information from the set of eigenvalues. Relying on the learned representations of eigenvalues, we also design a decoder with a bank of learnable bases. Finally, by combining these bases, Specformer can construct a permutation-equivariant and non-local graph convolution. In summary, our contributions are as follows:

- We propose a novel Transformer-based set-to-set spectral filter along with learnable bases, called Specformer, which effectively captures both magnitudes and relative differences of all eigenvalues of the graph Laplacian.

- We show that Specformer is permutation equivariant and can perform non-local graph convolutions, which is non-trivial to achieve in many spatial GNNs.

- Experiments on synthetic datasets show that Specformer learns to better recover the given spectral filters than other spectral GNNs.

- Extensive experiments on various node-level and graph-level benchmarks demonstrate that Specformer outperforms state-of-the-art GNNs and learns meaningful spectrum patterns.

## 2 RELATED WORK

Existing GNNs can be roughly divided into two categories: spatial and spectral GNNs.

**Spatial GNNs.** Spatial GNNs like GAT (Velickovic et al., 2018) and MPNN (Gilmer et al., 2017) leverage message passing to aggregate local information from neighborhoods. By stacking multiple layers, spatial GNNs can possibly learn long-range dependencies but suffer from over-smoothing (Oono & Suzuki, 2020) and over-squashing (Topping et al., 2022). Therefore, how to balance local and global information is an important research topic for spatial GNNs. We refer readers to (Wu et al., 2021; Zhou et al., 2020; Liao, 2021) for a more detailed discussion about spatial GNNs.

**Spectral GNNs.** Spectral GNNs (Ortega et al., 2018; Dong et al., 2020; Wu et al., 2019; Zhu et al., 2021; Bo et al., 2021; Chang et al., 2021; Yang et al., 2022) leverage the spectrum of graph Laplacian to perform convolutions in the spectral domain. A popular subclass of spectral GNNs leverages different kinds of orthogonal polynomials to approximate arbitrary filters, including Monomial (Chien et al., 2021), Chebyshev (Defferrard et al., 2016; Kipf & Welling, 2017; He et al., 2022), Bernstein (He et al., 2021), and Jacobi (Wang & Zhang, 2022). Relying on the diagonalization of symmetric matrices, they avoid direct spectral decomposition and guarantee localization. However, all such polynomial filters are scalar-to-scalar functions, and the bases are pre-defined, which limits their expressiveness. Another subclass requires either full or partial spectral decomposition, such as SpectralCNN (Estrach et al., 2014) and LanczosNet (Liao et al., 2019). They parameterize the spectral filters by neural networks, thus being more expressive than truncated polynomials. However, such spectral filters are still limited as they do not capture the dependencies among multiple eigenvalues.

**Graph Transformer.** Transformers and GNNs are closely relevant since the attention weights of Transformer can be seen as a weighted adjacency matrix of a fully connected graph. Graph Transformers Dwivedi & Bresson (2020) combine both and have gained popularity recently. Graphormer (Ying et al., 2022), SAN (Kreuzer et al., 2021), and GPS (Rampásek et al., 2022) design powerful positional and structural embeddings to further improve their expressive power. Graph Transformers still belong to spatial GNNs, although the high-cost self-attention is non-local. The limitation of spatial attention compared to spectral attention has been discussed in (Bastos et al., 2022).

## 3 BACKGROUND

In this section, we introduce some preliminaries of graph signal processing (GSP) (Ortega et al., 2018) and Transformer (Vaswani et al., 2017a).

**Preliminary.** Assume that we have a graph $\mathcal{G} = (\mathcal{V}, \mathcal{E})$, where $\mathcal{V}$ denotes the node set with $|\mathcal{V}| = n$ and $\mathcal{E}$ is the edge set. The corresponding adjacency matrix $\boldsymbol{A} \in \{0,1\}^{n \times n}$, where $A_{ij} = 1$ if there is an edge between nodes $i$ and $j$, and $A_{ij} = 0$ otherwise. The normalized graph Laplacian matrix is defined as $\boldsymbol{L} = \boldsymbol{I}_n - \boldsymbol{D}^{-1/2} \boldsymbol{A} \boldsymbol{D}^{-1/2}$, where $\boldsymbol{I}_n$ is the $n \times n$ identity matrix and $\boldsymbol{D}$ is the diagonal degree matrix with diagonal entries $D_{ii} = \sum_j A_{ij}$ for all $i \in \mathcal{V}$ and off-diagonal entries $D_{ij} = 0$ for $i \neq j$. We assume $\mathcal{G}$ is undirected. Hence, $\boldsymbol{L}$ is a real symmetric matrix, whose spectral decomposition can be written as $\boldsymbol{L} = \boldsymbol{U} \boldsymbol{\Lambda} \boldsymbol{U}^\top$, where the columns of $\boldsymbol{U}$ are the eigenvectors and $\boldsymbol{\Lambda} = \text{diag}([\lambda_1, \lambda_2, \ldots, \lambda_n])$ are the corresponding eigenvalues ranged in $[0, 2]$.

**Graph Signal Processing (GSP).** Spectral GNNs rely on several important concepts from GSP, namely, spectral filtering, graph Fourier transform and its inverse. The graph Fourier transform is written as $\hat{\boldsymbol{x}} = \boldsymbol{U}^\top \boldsymbol{x}$, where $\boldsymbol{x} \in \mathbb{R}^{n \times 1}$ is a graph signal and $\hat{\boldsymbol{x}} \in \mathbb{R}^{n \times 1}$ represents the Fourier coefficients. Then a spectral filter $G_\theta$ is used to scale $\hat{\boldsymbol{x}}$. Finally, the inverse graph Fourier transform is applied to yield the filtered signal in spatial domain $\tilde{\boldsymbol{x}} = \boldsymbol{U} G_\theta \hat{\boldsymbol{x}}$. The key task in GSP is to design a powerful spectral filter $G_\theta$ so that we can exploit the useful frequency information.

**Transformer.** Transformer is a powerful deep learning model, which is widely used in natural language processing (Devlin et al., 2019), vision (Dosovitskiy et al., 2021), and graphs (Ying et al., 2022; Rampásek et al., 2022). Each Transformer layer consists of two components: a multi-head self-attention (MHA) module and a token-wise feed-forward network (FFN). Given the input representations $\boldsymbol{H} = [\boldsymbol{h}_1^\top, \ldots, \boldsymbol{h}_n^\top] \in \mathbb{R}^{n \times d}$, where $d$ is the hidden dimension, MHA first projects $\boldsymbol{H}$ into query, key and value through three matrices ($\boldsymbol{W}^Q$, $\boldsymbol{W}^K$ and $\boldsymbol{W}^V$) to calculate attentions. And FFN is then used to add transformation. The model can be written as follows where we denote the query dimension as $d_q$. For simplicity, we omit the bias and the description of multi-head attention.

$$\text{Attention}(\boldsymbol{Q}, \boldsymbol{K}, \boldsymbol{V}) = \text{Softmax}(\frac{\boldsymbol{Q}\boldsymbol{K}^\top}{\sqrt{d_q}})\boldsymbol{V},$$
$$\boldsymbol{Q} = \boldsymbol{H}\boldsymbol{W}^Q, \ \boldsymbol{K} = \boldsymbol{H}\boldsymbol{W}^K, \ \boldsymbol{V} = \boldsymbol{H}\boldsymbol{W}^V. \tag{1}$$

## 4 SPECFORMER

In this section, we introduce our Specformer model. Fig. 1 illustrates the model architecture. We first explain how we encode the eigenvalues and use Transformer to capture their dependencies to yield useful representations. Then we turn to the decoder, which learns new eigenvalues from the representations and reconstructs the graph Laplacian matrix for graph convolution. Finally, we discuss the relationship between our Specformer and other methods, including MPNNs, polynomial GNNs, and graph Transformers.

### 4.1 EIGENVALUE ENCODING

We design a powerful set-to-set spectral filter using Transformer to leverage both magnitudes and relative differences of all eigenvalues. However, the expressiveness of self-attention will be restricted heavily if we directly use the scalar eigenvalues to calculate the attention maps. Therefore, it is important to find a suitable function, $\rho(\lambda) : \mathbb{R}^1 \to \mathbb{R}^d$, to map each eigenvalue from a scalar to a meaningful vector. We use an eigenvalue encoding function as follows.

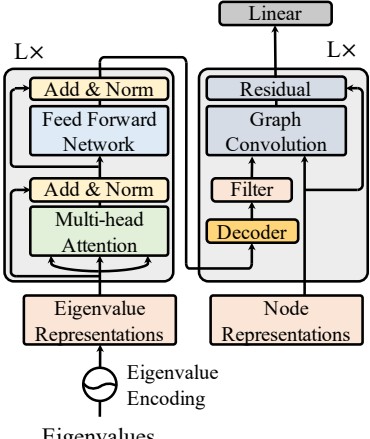

Figure 1: Illustration of the proposed Specformer.

$$\rho(\lambda, 2i) = \sin(\epsilon\lambda/10000^{2i/d}),$$
$$\rho(\lambda, 2i+1) = \cos(\epsilon\lambda/10000^{2i/d}), \tag{2}$$

where $i$ is the dimension of the representations and $\epsilon$ is a hyperparameter. The benefits of $\rho(\lambda)$ are three-fold: (1) It can capture the relative frequency shifts of eigenvalues and provides high-dimension

vector representations. (2) It has the wavelengths from $2\pi$ to $10000 \cdot 2\pi$, which forms a multi-scale representation for eigenvalues. (3) It can control the influence of $\lambda$ by adjusting the value of $\epsilon$.

The choice of $\epsilon$ is crucial because we find that only the first few dimensions of $\rho(\lambda)$ can distinguish different eigenvalues if we simply set $\epsilon = 1$. The reason is that eigenvalues lie in the range $[0, 2]$, and the value of $\lambda / 10000^{2i/d}$ will change slightly when $i$ becomes larger. Therefore, it is important to assign a large value of $\epsilon$ to enlarge the influence of $\lambda$. Experiments can be seen in Appendix C.1.

Notably, although the eigenvalue encoding (EE) is similar to the positional encoding (PE) of Transformer, they act quite differently. PE describes the information of discrete positions in the spatial domain. While EE represents the information of continuous eigenvalues in the spectral domain. Applying PE to the spatial positions (*i.e.*, indices) of eigenvalues will destroy the permutation equivariance property, thereby impairing the learning ability.

The initial representations of eigenvalues are the concatenation of eigenvalues and their encodings, $\boldsymbol{Z} = [\lambda_1 || \rho(\lambda_1), \cdots, \lambda_n || \rho(\lambda_n)]^\top \in \mathbb{R}^{n \times (d+1)}$. Then a standard Transformer block is used to learn the dependency between eigenvalues. We first apply layer normalization (LN) on the representations before feeding them into other sub-layers, *i.e.*, MHA and FFN. This *pre-norm* trick has been used widely to improve the optimization of Transformer (Ying et al., 2022):

$$\begin{aligned} \tilde{\boldsymbol{Z}} &= \text{MHA}(\text{LN}(\boldsymbol{Z})) + \boldsymbol{Z}, \\ \hat{\boldsymbol{Z}} &= \text{FFN}(\text{LN}(\tilde{\boldsymbol{Z}})) + \tilde{\boldsymbol{Z}}. \end{aligned} \tag{3}$$

After stacking multiple Transformer blocks, we obtain the expressive representations of the spectrum.

## 4.2 EIGENVALUE DECODING

Based on the representations returned by the encoder, the decoder can learn new eigenvalues for spectral filtering. Recent studies (Yang et al., 2022; Wang & Zhang, 2022) show that assigning each feature dimension a separate spectral filter improves the performance of GNNs. Motivated by this discovery, our decoder first decodes several bases. An FFN is then used to combine these bases to construct the final graph convolution.

**Spectral filters.** In general, the bases should learn to cover different information of the graph signal space as much as possible. For this purpose, we utilize the multi-head attention mechanism because each head has its own self-attention module. Specifically, the representations learned by each head will be fed into the decoder to perform spectral filtering to get the new eigenvalues.

$$\boldsymbol{Z}_m = \text{Attention}(\boldsymbol{Q}\boldsymbol{W}_m^Q, \boldsymbol{K}\boldsymbol{W}_m^K, \boldsymbol{V}\boldsymbol{W}_m^V), \quad \boldsymbol{\lambda}_m = \phi(\boldsymbol{Z}_m \boldsymbol{W}_\lambda), \tag{4}$$

where $\boldsymbol{Z}_m$ denotes the representations learned by the $m$-th heads, and $\phi$ is the activation, *e.g.*, ReLU or Tanh, which is optional. $\boldsymbol{\lambda}_m \in \mathbb{R}^{n \times 1}$ is the $m$-th eigenvalues after the spectral filtering.

**Learnable bases.** After get $M$ filtered eigenvalues, we use a FFN: $\mathbb{R}^{M+1} \to \mathbb{R}^d$ to construct the learnable bases. We first reconstruct individual new bases, concatenate them along the channel dimension, and feed them to a FFN as below,

$$\boldsymbol{S}_m = \boldsymbol{U}\text{diag}(\boldsymbol{\lambda}_m)\boldsymbol{U}^\top, \quad \hat{\boldsymbol{S}} = \text{FFN}([\boldsymbol{I}_n || \boldsymbol{S}_1 || \cdots || \boldsymbol{S}_M]), \tag{5}$$

where $\boldsymbol{S}_m \in \mathbb{R}^{n \times n}$ is the $m$-th new basis and $\hat{\boldsymbol{S}} \in \mathbb{R}^{n \times n \times d}$ is the combined version. Note that our bases here serve similar purpose as those polynomial bases in the literature. But the way they are combined is learned rather than following certain recursions as in Chebyshev polynomials.

We have three optional ways to leverage this design of new bases. (1) Shared filters and shared FFN. This model has the least parameters, where the basis $\hat{\boldsymbol{S}}$ is shared across all graph convolutional layers. (2) Shared filters and layer-specific FFN, which compromises between parameters and performance, *e.g.*, $\hat{\boldsymbol{S}}^{(l)} = \text{FFN}^{(l)}([\boldsymbol{I}_n || \boldsymbol{S}_1 || \cdots || \boldsymbol{S}_M])$ where the superscript $l$ denotes the index of layer. (3) Layer-specific filters and layer-specific FFN. This model has the most parameters and each layer has its own encoder and decoder, *e.g.*, $\hat{\boldsymbol{S}}^{(l)} = \text{FFN}^{(l)}([\boldsymbol{I}_n || \boldsymbol{S}_1^{(l)} || \cdots || \boldsymbol{S}_M^{(l)}])$. We refer these three models as Specformer-Small, Specformer-Medium, and Specformer-Large.

### 4.3 GRAPH CONVOLUTION

Finally, we assign each feature dimension a separate graph Laplacian matrix based on the learned basis $\hat{S}$, which can be written as follows:

$$\hat{X}_{:,i}^{(l-1)} = \hat{S}_{:,:,i}X_{:,i}^{(l-1)}, \quad X^{(l)} = \sigma\left(\hat{X}^{(l-1)}W_x^{(l-1)}\right) + X^{(l-1)}, \tag{6}$$

where $X^{(l)}$ is the node representations in the $l$-th layer, $\hat{X}_{:,i}^{(l-1)}$ is the $i$-th channel dimension, $W_x^{(l-1)}$ is the transformation, and $\sigma$ is the activation. The residual connection is an optional choice. By stacking multiple graph convolutional layers, Specformer can effectively learn node representations.

### 4.4 KEY PROPERTIES COMPARED TO RELATED MODELS

**Specformer v.s. Polynomial GNNs.** Specformer replaces the fixed bases of polynomials, *e.g.*, $\lambda, \lambda^2, \cdots, \lambda^k$, with learnable bases, which has two major advantages: (1) Universality. Polynomial GNNs are the special cases of Specformer because the learnable bases can approximate any polynomials. (2) Flexibility. Polynomial GNNs are designed to learn a shared function for all eigenvalues whereas Specformer can learn eigenvalue-specific functions, thus being more flexible.

**Specformer v.s. MPNNs.** MPNNs aggregate the local information from neighborhood one hop per layer. This localization capability enables MPNNs with high computational efficiency but weakens the ability in capturing the global information. Specformer is inherently non-local due to the use of (often dense) eigenvectors. The learned graph Laplacian $UG_\theta U^\top = \theta_1 u_1 u_1^\top + \cdots + \theta_n u_n u_n^\top$. Because $u_i$ is a eigenvector, $u_i u_i^\top$ constructs a fully-connected graph. Therefore, our Specformer can break the localization of MPNNs and leverage global information.

**Specformer v.s. Graph Transformers.** Graphormer (Ying et al., 2022) has shown that graph Transformer can perform well on graph-level tasks. However, existing graph Transformers do not show competitiveness in the node-level tasks, *e.g.*, node classification. Recent studies (Bastos et al., 2022; Wang et al., 2022; Shi et al., 2022) provide some evidence for this phenomenon. They show that Transformer is essentially a low-pass filter. Therefore, graph Transformers cannot handle the complex node label distribution, *e.g.*, homophilic and heterophilic. On the contrary, as we will see in the experiment section, Specformer can learn arbitrary bases for graph convolution and perform well on both node-level and graph-level tasks.

Besides the above advantages, we show that our Specformer has the following theoretical properties.
**Proposition 1.** *Specformer is permutation equivariant.*
**Proposition 2.** *Specformer can approximate any univariate and multivariate continuous functions.*

Proposition 1 shows that Specformer can learn permutation-equivariant node representations. Proposition 2 states that Specformer is more expressive than other graph filters. First, the ability to approximate any univariate functions generalizes existing scalar-to-scalar filters. Besides, Specformer can handle multiple eigenvalues and learn multivariate functions, so it can approximate a broader range of filter functions than scalar-to-scalar filters. All proofs are provided in Appendix D

**Complexity.** Specformer has two parts of computation: spectral decomposition and forward process. Spectral decomposition is pre-computed and has the complexity of $\mathcal{O}(n^3)$. The forward complexity has three parts: Transformer, learnable bases, and graph convolution. Their corresponding complexities are $\mathcal{O}(n^2 d + nd^2)$, $\mathcal{O}(Mn^2)$ and $\mathcal{O}(Lnd)$, respectively, where $n, M, L$ represent the number of nodes, filters, and layers, and $d$ is the hidden dimension. The overall forward complexity is $\mathcal{O}(n^2(d+M) + nd(L+d))$. The overall complexity of Specformer is the sum of the forward complexity and the decomposition complexity amortized over the number of uses in training and inference, rather than a simple summation of the two. See Appendix C.2 for more discussion.

**Scalability.** When applying Specformer to large graphs, one can use the Sparse Generalized Eigenvalue (SGE) algorithms (Cai et al., 2021) to calculate $q$ eigenvalues and eigenvectors, in which case the forward complexity will reduce to $(q^2(d+M) + nd(L+d))$.

## 5 EXPERIMENTS

In this section, we conduct experiments on a synthetic dataset and a wide range of real-world graph datasets to verify the effectiveness of our Specformer.

Table 1: Node regression results, mean of the sum of squared error & $R^2$ score, on synthetic data.

| Model ($\sim$2k param.) | Low-pass $\exp(-10\lambda^2)$ | High-pass $1 - \exp(-10\lambda^2)$ | Band-pass $\exp(-10(\lambda-1)^2)$ | Band-rejection $1 - \exp(-10(\lambda-1)^2)$ | Comb $|\sin(\pi\lambda)|$ |
|---|---|---|---|---|---|
| GCN | 3.4799(.9872) | 67.6635(.2364) | 25.8755(.1148) | 21.0747(.9438) | 50.5120(.2977) |
| GAT | 2.3574(.9905) | 21.9618(.7529) | 14.4326(.4823) | 12.6384(.9652) | 23.1813(.6957) |
| ChebyNet | 0.8220(.9973) | 0.7867(.9903) | 2.2722(.9104) | 2.5296(.9934) | 4.0735(.9447) |
| GPR-GNN | 0.4169(.9984) | 0.0943(.9986) | 3.5121(.8551) | 3.7917(.9905) | 4.6549(.9311) |
| BernNet | 0.0314(.9999) | 0.0113(.9999) | 0.0411(.9984) | 0.9313(.9973) | 0.9982(.9868) |
| JacobiConv | 0.0003(.9999) | 0.0064(.9999) | 0.0213(.9999) | 0.0156(.9999) | 0.2933(.9995) |
| Specformer | **0.0002(.9999)** | **0.0026(.9999)** | **0.0017(.9999)** | **0.0014(.9999)** | **0.0057(.9999)** |

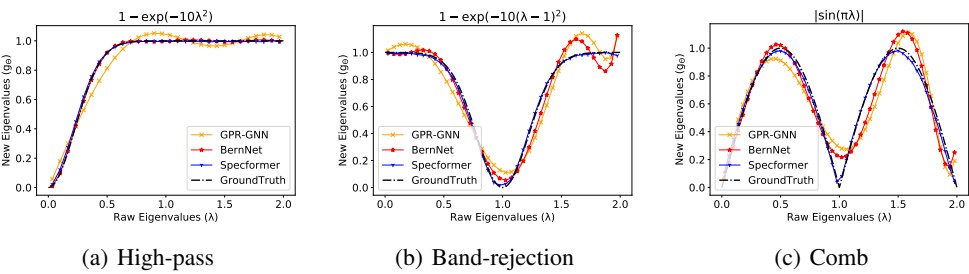

(a) High-pass  (b) Band-rejection  (c) Comb

Figure 2: Illustrations of filters learned by two polynomial GNNs and Specformer.

## 5.1 LEARNING SPECTRAL FILTERS ON SYNTHETIC DATA

**Dataset description.** We take 50 images with the resolution of $100 \times 100$ from the Image Processing Toolbox [1]. Each image is processed as a 2D regular 4-neighborhood grid graph, and the values of pixels are the node features. Therefore, these images share the same adjacency matrix $\boldsymbol{A} \in \mathbb{R}^{10000 \times 10000}$ and the $m$-th image has its graph signal $\boldsymbol{x}_m \in \mathbb{R}^{10000 \times 1}$. Five predefined graph filters are used to generate ground truth graph signals. For example, if we use the low-pass filter with $G_\theta = \exp(-10\lambda^2)$, the filtered graph signal is calculated by $\tilde{\boldsymbol{x}}_m = \boldsymbol{U}\text{diag}[\exp(-10\lambda_1^2), \ldots, \exp(-10\lambda_n^2)]\boldsymbol{U}^\top \boldsymbol{x}_m$.

**Setup.** We choose six Spectral GNNs as baselines: GCN (Kipf & Welling, 2017), GAT (Velickovic et al., 2018), ChebyNet (Defferrard et al., 2016), GPR-GNN (Chien et al., 2021), BernNet (He et al., 2021), and JacobiConv (Wang & Zhang, 2022). Each method takes $\boldsymbol{A}$ and $\boldsymbol{x}_m$ as inputs, and tries to minimize the sum of squared error between the outputs $\hat{\boldsymbol{x}}_m$ and the pre-filtered graph signal $\tilde{\boldsymbol{x}}_m$. We tune the number of hidden units to ensure that each method has nearly 2K trainable parameters. The polynomial order is set to 10 for ChebyNet, GPR-GNN, and BernNet. For our model, we use Specformer-Small with 16 hidden units and 1 head. In training, the maximum number of epochs is set to 2000, and the model will be stopped early if the loss does not descend 200 epochs. All regularization tricks are removed. The learning rate is set to 0.01 for all models. We use two metrics to evaluate each method: sum of squared error and $R^2$ score.

**Results.** The quantitative experiment results are shown in Table 1, from which we can see that Specformer achieves the best performance on all synthetic graphs. Especially, it has more improvements on challenging graphs, such as Band-rejection and Comb. This validates the effectiveness of Specformer in learning complex graph filters. In addition, we can see that GCN and GAT only perform better on the homophilic graph, which reflects that only using low-frequency information is not enough. Polynomial-based GNNs, *i.e.*, ChebyNet, GPR-GNN, BernNet, and JacobiConv, have more stable performances. But their expressiveness is still weaker than Specformer. We visualize the graph filters learned by GPR-GNN, BernNet, and Specformer in Figure 2, which further validates our claims. The horizontal axis presents the original eigenvalues, and the vertical axis indicates the corresponding new eigenvalues. For clarity, we uniformly downsample the eigenvalues at a ratio of 1:200 and only visualize three graphs because the situations of Low-pass and High-pass are similar. The same goes for Band-pass and Band-rejection. It can be seen that all methods can fit the easy filters well, *i.e.*, High-pass. However, the polynomial-based GNNs cannot learn the narrow bands in Band-rejection and Comb, *e.g.*, $\lambda \in [0.75, 1.25]$, which harms their performance. On the contrary, Specformer fits the ground truth precisely, reflecting the superior learning ability over polynomials. The spatial results, *i.e.*, filtered images, can be seen in Appendix C.3.

---

[1] https://ww2.mathworks.cn/products/image.html

Table 2: Results on real-world node classification tasks. Mean accuracy (%) ± 95% confidence interval. * means re-implemented baselines. "OOM" means out of GPU memory.

| | Param. on Photo | Heterophilic | | | | Homophilic | | | |
|---|---|---|---|---|---|---|---|---|---|
| | | Chameleon | Squirrel | Actor | Penn94 | Cora | Citeseer | Photo | arXiv |
| Spatial-based GNNs | | | | | | | | | |
| GCN | 48K | 59.61±2.21 | 46.78±0.87 | 33.23±1.16 | 82.47±0.27 | 87.14±1.01 | 79.86±0.67 | 88.26±0.73 | 71.74±0.29 |
| GAT | 49K | 63.13±1.93 | 44.49±0.88 | 33.93±2.47 | 81.53±0.55 | 88.03±0.79 | 80.52±0.71 | 90.94±0.68 | 71.82±0.23 |
| $H_2$GCN | 60K | 57.11±1.58 | 36.42±1.89 | 35.86±1.03 | OOM | 86.92±1.37 | 77.07±1.64 | 93.02±0.91 | OOM |
| GCNII | 49K | 63.44±0.85 | 41.96±1.02 | 36.89±0.95 | 82.92±0.59 | 88.46±0.82 | 79.97±0.65 | 89.94±0.31 | 72.04±0.19 |
| Spectral-based GNNs | | | | | | | | | |
| LanczosNet* | 50K | 64.81±1.56 | 48.64±1.77 | 38.16±0.91 | 81.55±0.26 | 87.77±1.45 | 80.05±1.65 | 93.21±0.85 | 71.46±0.39 |
| ChebyNet | 48K | 59.28±1.25 | 40.55±0.42 | 37.61±0.89 | 81.09±0.33 | 86.67±0.82 | 79.11±0.75 | 93.77±0.32 | 71.12±0.22 |
| GPR-GNN | 48K | 67.28±1.09 | 50.15±1.92 | 39.92±0.67 | 81.38±0.16 | 88.57±0.69 | 80.12±0.83 | 93.85±0.28 | 71.78±0.18 |
| BernNet | 48K | 68.29±1.58 | 51.35±0.73 | 41.79±1.01 | 82.47±0.21 | 88.52±0.95 | 80.09±0.79 | 93.63±0.35 | 71.96±0.27 |
| ChebNetII | 48K | 71.37±1.01 | 57.72±0.59 | 41.75±1.07 | 83.12±0.22 | 88.71±0.93 | 80.53±0.79 | 94.92±0.33 | 72.32±0.23 |
| JacobiConv | 48K | 74.20±1.03 | 57.38±1.25 | 41.17±0.64 | 83.35±0.11 | **88.98±0.46** | 80.78±0.79 | 95.43±0.23 | 72.14±0.17 |
| Graph Transformers | | | | | | | | | |
| Transformer* | 37K | 46.39±1.97 | 31.90±3.16 | 39.95±1.64 | OOM | 71.83±1.68 | 70.55±1.20 | 90.05±1.50 | OOM |
| Graphormer* | 139K | 54.49±3.11 | 36.96±1.75 | 38.45±1.38 | OOM | 67.71±0.78 | 73.30±1.21 | 85.20±4.12 | OOM |
| Specformer | 32K | **74.72±1.29** | **64.64±0.81** | **41.93±1.04** | **84.32±0.32** | 88.57±1.01 | **81.49±0.94** | **95.48±0.32** | **72.37±0.18** |

## 5.2 NODE CLASSIFICATION

**Datasets.** For the node classification task, we perform experiments on four homophilic datasets, *i.e.*, Cora, Citeser, Amazon-Photo and ogbn-arXiv, and four heterophilic datasets, *i.e.*, Chameleon, Squirrel, Actor and Penn94. Penn94 (Lim et al., 2021) and arXiv (Hu et al., 2020) are two large scale datasets. Other datasets, provided by (Rozemberczki et al., 2021; Pei et al., 2020), are commonly used to evaluate the performance of GNNs on heterophilic and homophilic graphs.

**Baselines and settings.** We benchmark our model against a series of competitive baselines, including spatial GNNs, spectral GNNs, and graph Transformers. For all datasets, we use the full-supervised split, *i.e.*, 60% for training, 20% for validation, and 20% for testing, as suggested in (He et al., 2021). All methods run 10 times and report the mean accuracy with a 95% confidence interval. For polynomial GNNs, we set the order of polynomials $K = 10$. For other methods, we use a 2-layer module. To ensure all models have similar numbers of parameters, in six small datasets, we set the hidden size $d = 64$ for spatial and spectral GNNs and $d = 32$ for graph Transformers and Specformer. The total numbers of parameters on Photo dataset are shown in Table 2. On two large datasets, we use truncated spectral decomposition to improve the scalability. Based on the filters learned on the small datasets, we find that band-rejection filters are important for heterophilic datasets and low-pass filters are suitable for homophilic datasets. See Figures 4(b) and 4(d). Therefore, we use eigenvectors with the smallest 3000 (low-frequency) and largest 3000 eigenvalues (high-frequency) for Penn94, and eigenvectors with the smallest 5000 eigenvalues (low-frequency) for arXiv. We use one layer for Specformer and set $d = 64$ in Penn94 and $d = 512$ in arXiv for all methods, as suggested by (Lim et al., 2021; He et al., 2022). More details, *e.g.*, optimizers, can be found in Appendix A.

**Results.** In Table 2, we can find that Specformer outperforms state-of-the-art baselines on 7 out of 8 datasets and achieves 12% relative improvement on the Squirrel dataset, which validates the superior learning ability of Specformer. In addition, the improvement is more pronounced on heterophilic datasets than on homophilic datasets. This is probably caused by the easier fitting of the low-pass filters in homophilic datasets. The same phenomenon is also observed in the synthetic graphs. An interesting observation is that the improvement on larger graphs, *e.g.*, Actor and Photo, is less than that on smaller graphs. One possible reason is that the role of the self-attention mechanism is weakened, *i.e.*, the attention values become uniform due to a large number of tokens. We notice that Specformer has a slightly higher variance than the baselines. This is because we set a large dropout rate to prevent overfitting. On the two large graph datasets, we can see that graph Transformers are memory-consuming due to the self-attention. On the contrary, Specformer reduces the time and space costs by using the truncated decomposition and shows better scalability than graph Transformers. The time and space overheads are listed in Appendix C.2.

## 5.3 GRAPH CLASSIFICATION AND REGRESSION

**Datasets.** We conduct experiments on three graph-level datasets with different scales. ZINC (Dwivedi et al., 2020) is a small subset of a large molecular dataset, which contains 12K graphs in

Table 3: Results on graph-level datasets. ↓ means lower the better, and ↑ means higher the better.

| Model | ZINC(↓) | MolHIV(↑) | MolPCBA(↑) |
|---|---|---|---|
| GCN | $0.367 \pm 0.011$ | $0.7599 \pm 0.0119$ | $0.2424 \pm 0.0034$ |
| GIN | $0.526 \pm 0.051$ | $0.7707 \pm 0.0149$ | $0.2703 \pm 0.0023$ |
| GatedGCN | $0.090 \pm 0.001$ | - | $0.267 \pm 0.002$ |
| CIN | $0.079 \pm 0.006$ | $\mathbf{0.8094 \pm 0.0057}$ | - |
| GIN-AK+ | $0.080 \pm 0.001$ | $0.7961 \pm 0.0119$ | $0.2930 \pm 0.0044$ |
| GSN | $0.101 \pm 0.010$ | $0.7799 \pm 0.0100$ | - |
| DGN | $0.168 \pm 0.003$ | $0.7970 \pm 0.0097$ | $0.2885 \pm 0.0030$ |
| PNA | $0.188 \pm 0.004$ | $0.7905 \pm 0.0132$ | $0.2838 \pm 0.0035$ |
| Spec-GN | $0.070 \pm 0.002$ | - | $0.2965 \pm 0.0028$ |
| SAN | $0.139 \pm 0.006$ | $0.7785 \pm 0.0025$ | $0.2765 \pm 0.0042$ |
| Graphormer[2] | $0.122 \pm 0.006$ | $0.7640 \pm 0.0022$ | $0.2643 \pm 0.0017$ |
| GPS | $0.070 \pm 0.004$ | $0.7880 \pm 0.0101$ | $0.2907 \pm 0.0028$ |
| Specformer | $\mathbf{0.066 \pm 0.003}$ | $0.7889 \pm 0.0124$ | $\mathbf{0.2972 \pm 0.0023}$ |

Table 4: Ablation studies on node-level and graph-level tasks.

| Encoder | | Decoder | | | Node-level | | Graph-level |
|---|---|---|---|---|---|---|---|
| $\rho(\lambda)$ | Attention | Small | Medium | Large | Squirrel (↑) | Citeseer (↑) | MolPCBA (↑) |
| | | | ✓ | | 33.05 | 80.57 | 0.2696 |
| ✓ | | | ✓ | | 63.78 | 81.17 | 0.2933 |
| ✓ | ✓ | | ✓ | | 64.64 | 81.49 | 0.2970 |
| ✓ | ✓ | ✓ | | | 64.51 | 81.47 | 0.2912 |
| ✓ | ✓ | | | ✓ | 65.10 | 80.00 | 0.2972 |

total. MolHIV and MolPCBA are taken from the Open Graph Benchmark (OGB) datasets (Hu et al., 2020). MolHIV is a medium dataset that has nearly 41K graphs. MolPCBA is the largest, containing 437K graphs. For all datasets, nodes represent the atoms, and edges indicate the bonds.

**Baselines and Settings.** We choose popular MPNNs (GCN, GIN, and GatedGNN), graph Transformers with positional or structural embedding (SAN, Graphormer, and GPS), and other state-of-the-art GNNs (CIN, GIN-AK+, etc.) as the baselines of graph-level tasks. For the ZINC dataset, we tune the hyperparameters of Specformer to ensure that the total parameters are around 500K.

**Results.** We apply Specformer-Small, Medium, and Large for ZINC, MolHIV, and MolPCBA, respectively. The results are shown in Table 3. It can be seen that Specformer outperforms the state-of-the-art models in ZINC and MolPCBA datasets, without using any hand-crafted features or pre-defined polynomials. This phenomenon proves that directly using neural networks to learn the graph spectrum is a promising way to construct powerful GNNs.

## 5.4 ABLATION STUDIES

We perform ablation studies on two node-level datasets and one graph-level dataset to evaluate the effectiveness of each component. The results are shown in Table 4. The top three lines show the effect of the encoder, *i.e.*, eigenvalue encoding (EE) and self-attention. It can be seen that EE is more important on Squirrel than on Citeseer. The reason is that the spectral filter of Squirrel is more difficult to learn. Therefore, the model needs the encoding to learn better representations. The attention module consistently improves performance by capturing the dependency among eigenvalues.

The bottom three lines verify the performance of graph filters at different scales. We can see that in the easy task, *e.g.*, Citeseer, the Small and Medium models have similar performance, but the Large model causes serious overfitting. In the hard task, *e.g.*, Squirrel and MolPCBA, the Large model is slightly better than the Medium model but outperforms the Small model a lot, implying that adding the number of parameters can boost the performance. In summary, it is important to consider the difficulty of tasks when selecting models.

---

[2]We retrain Graphomer on MolHIV and MolPCBA datasets without using pre-training and augmentation.

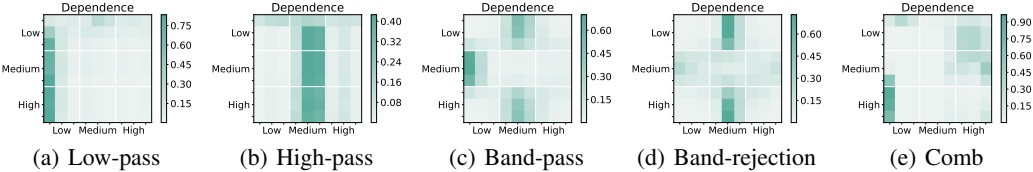

Figure 3: The dependency of eigenvalues on synthetic graphs.

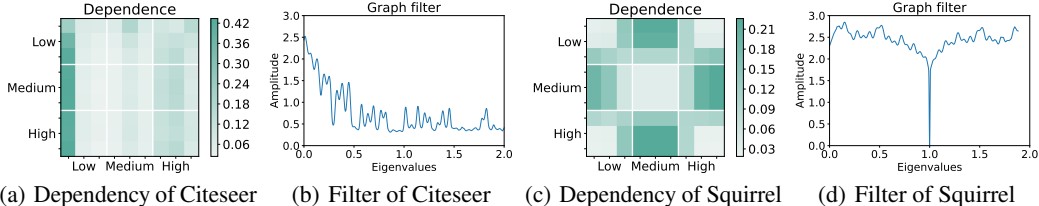

Figure 4: The dependency and learned filters of heterophilic and homophilic datasets.

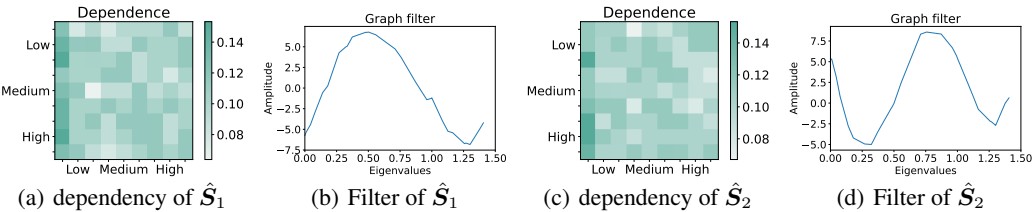

Figure 5: The dependency and basic filters of ZINC dataset.

## 5.5 VISUALIZATIONS

We now investigate the dependency of eigenvalues and filters learned by Specformer. To make the dependency clearer, we first quantize the self-attention weight matrix by grouping the eigenvalues into three frequency bands: Low $\in [0, \frac{2}{3})$, Medium $\in [\frac{2}{3}, \frac{4}{3})$, and High $\in [\frac{4}{3}, 2]$. We then compute the quantized dependency. More details are given in Appendix B.2.

The results are shown in Figures 3, 4, and 5, from which we have some interesting observations. (1) Similar dependency patterns can be learned on different graphs. In low-pass filtering, *e.g.*, Citeseer and Low-pass, all frequency bands tend to use the low-frequency information. While, in the band-related filtering, *e.g.*, Squirrel, Band-pass, and Band-rejection, the low- and high-frequency highly depend on the medium-frequency, and the situation of the medium-frequency is opposite. (2) The more difficult the task, the less obvious the dependency. On Comb and ZINC, the dependency of eigenvalues is inconspicuous. (3) On graph-level datasets, the decoder can learn different filters. Figure 5 shows two basic filters. It can be seen that $\hat{S}_1$ and $\hat{S}_2$ have different dependencies and patterns, which are different from node-level tasks, where only one filter is needed. The finding suggests that these graph-level tasks are more difficult than node-level tasks and are still challenging for spectral GNNs.

## 6 CONCLUSION

In this paper, we propose Specformer that leverages Transformer to build a set-to-set spectral filter along with learnable bases. Specformer effectively captures magnitudes and relative dependencies of the eigenvalues in a permutation-equivariant fashion and can perform non-local graph convolution. Experiments on synthetic and real-world datasets demonstrate that Specformer outperforms various GNNs and learns meaningful spectrum patterns. A promising future direction is to improve the efficiency of Specformer through sparsifying the self-attention matrix of Transformer.

ACKNOWLEDGMENTS

This work is supported in part by the National Natural Science Foundation of China (No. U20B2045, 62192784, 62172052, 62002029, 62172052, U1936014), BUPT Excellent Ph.D. Students Foundation (No. CX2022310), the NSERC Discovery Grants (No. RGPIN-2019-05448, No. RGPIN-2022-04636), and the NSERC Collaborative Research and Development Grant (No. CRDPJ 543676-19). Resources used in preparing this research were provided, in part, by Advanced Research Computing at the University of British Columbia, the Oracle for Research program, and Compute Canada.

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

# A  EXPERIMENTAL DETAILS

## A.1  DATASETS

Table 5: Detailed information of node-level datasets.

|            | Graphs | Nodes   | Edges     | Features | Classes |
|------------|--------|---------|-----------|----------|---------|
| Chameleon  | 1      | 2,277   | 36,101    | 2,325    | 5       |
| Squirrel   | 1      | 5,201   | 217,073   | 2,089    | 5       |
| Actor      | 1      | 7,600   | 33,544    | 932      | 5       |
| Cora       | 1      | 2,708   | 5,429     | 1,433    | 7       |
| Citeseer   | 1      | 3,327   | 4,732     | 3,703    | 6       |
| Photo      | 1      | 7,650   | 110,081   | 745      | 8       |
| Penn94     | 1      | 41,554  | 1,362,229 | 4,814    | 2       |
| arXiv      | 1      | 169,343 | 1,116,243 | 128      | 40      |

Table 6: Detailed information of graph-level datasets.

|          | # Graphs  | Avg. # nodes | Avg. # edges | Min # nodes | Max # nodes | Tasks          | Metric |
|----------|-----------|--------------|--------------|-------------|-------------|----------------|--------|
| ZINC     | 12,000    | 23.2         | 24.9         | 9           | 37          | Regression     | MAE    |
| MolHIV   | 41,127    | 25.5         | 27.5         | 2           | 222         | Classification | AUROC  |
| MolPCBA  | 437,929   | 26.0         | 28.1         | 1           | 332         | Classification | AP     |
| PCQM4Mv2 | 3,746,620 | 14.1         | 14.6         | 1           | 20          | Regression     | MAE    |

## A.2  DETAILED EXPERIMENTAL SETUP

**Data splitting.**   In the node classification task, Lim et al. (2021) provides five official splits for Penn94 dataset, Hu et al. (2020) provides one time-based split for arXiv dataset. Therefore, we run the Penn94 dataset five times, each with a different split. And we run the arXiv dataset ten times, each with the same split and a different initialization. For other datasets, we run the experiments ten times, each with a different split and initialization because there is no official splitting. In the graph-level tasks, we use the official splitting provided by Hu et al. (2020) and run the experiments ten times, each with a different initialization.

**Optimizer.**   For the node classification task, we use the Adam (Kingma & Ba, 2015) optimizer, as suggested by He et al. (2021); Wang & Zhang (2022). For graph-level tasks, we use the AdamW (Loshchilov & Hutter, 2019) optimizer, with the default parameters of $\epsilon =$ 1e-8 and $(\beta_1, \beta_2) = (0.99, 0.999)$, as suggested by Ying et al. (2022); Rampásek et al. (2022). Besides, we also use a learning rate scheduler for graph-level tasks, which is first a linear warm-up stage followed by a cosine decay.

**Model selection.**   In the node classification task, we run the experiments with 2000 epochs and stop the training in advance if the validation loss does not continuously decrease for 200 epochs. In the graph-level tasks, we run the experiments without early stop. Then we choose the model checkpoint with the lowest validation loss for evaluation.

**Environment.**   The environment in which we run experiments is:

- Operating system: Linux version 3.10.0-693.el7.x86_64

- CPU information: Intel(R) Xeon(R) Silver 4210 CPU @ 2.20GHz

- GeForce RTX 3090 (24GB)

**Hyperparameters.**   The hyperparameters of specformer can be seen in Tables 7 and 8.

Table 7: Hyperparameters of node-level datasets.

| Hyperparameter | Synthetic | Chameleon | Squirrel | Actor | Cora | Citeseer | Photo | Penn94 | arXiv |
|---|---|---|---|---|---|---|---|---|---|
| Layer | 1 | 2 | 2 | 2 | 2 | 2 | 2 | 1 | 1 |
| Heads | 1 | 4 | 2 | 1 | 2 | 2 | 4 | 1 | 1 |
| Hidden dim | 16 | 32 | 32 | 32 | 32 | 32 | 32 | 64 | 512 |
| Epoch | 2000 | 2000 | 2000 | 2000 | 2000 | 2000 | 2000 | 2000 | 2000 |
| Learning rate | 0.01 | 1e-3 | 1e-3 | 2e-4 | 2e-4 | 2e-4 | 2e-4 | 1e-3 | 1e-3 |
| Weight decay | 0 | 5e-4 | 1e-3 | 1e-4 | 1e-4 | 1e-3 | 1e-4 | 1e-3 | 0 |
| Transformer dropout | 0 | 0.2 | 0.1 | 0.5 | 0.2 | 0 | 0.2 | 0.0 | 0.1 |
| Feature dropout | 0 | 0.4 | 0.4 | 0.8 | 0.6 | 0.7 | 0.3 | 0.4 | 0.1 |
| Propagation dropout | 0 | 0.5 | 0.4 | 0.5 | 0.2 | 0.5 | 0.2 | 0.4 | 0.1 |
| Combination | Small | Medium | Medium | Medium | Medium | Medium | Medium | Medium | Medium |
| Parameters | 2,084 | 82,445 | 74,787 | 37,680 | 53,885 | 126,840 | 32,044 | 338,179 | 1,931,305 |

Table 8: Hyperparameters of graph-level datasets.

| Hyperparameter | ZINC | MolHIV | MolPCBA | PCQM4Mv2 |
|---|---|---|---|---|
| Layer | 4 | 8 | 8 | 8 |
| Heads | 8 | 4 | 8 | 8 |
| Hidden dim | 160 | 80 | 256 | 320 |
| Graph pooling | mean | mean | mean | mean |
| Epoch | 1000 | 50 | 30 | 150 |
| Warmup | 50 | 5 | 5 | 10 |
| Batch size | 32 | 64 | 64 | 256 |
| Learning rate | 1e-3 | 1e-4 | 5e-4 | 1e-4 |
| Weight decay | 5e-4 | 1e-4 | 5e-3 | 0 |
| Transformer dropout | 0.1 | 0.1 | 0.3 | 0.1 |
| Feature dropout | 0.05 | 0.1 | 0.1 | 0.1 |
| Propagation dropout | 0 | 0.3 | 0.1 | 0.1 |
| Combination | Small | Medium | Large | Medium |
| Parameter | 529,609 | 226,645 | 3,025,048 | 4,117,129 |

# B  IMPLEMENTATION DETAILS

## B.1  SPECFORMER LAYER

Here we explain how to incorporate the Specformer layer with edge features. Specifically, we first broadcast the node features to the edges and filter the mixed edge features. Finally, the filtered edge features are aggregated to yield new node features.

$$
\begin{aligned}
\boldsymbol{E} &= \boldsymbol{H}.\text{unsqueeze}(0) + \boldsymbol{E}, \\
\hat{\boldsymbol{E}} &= \boldsymbol{S} \odot \boldsymbol{E}, \\
\hat{\boldsymbol{H}} &= \hat{\boldsymbol{E}}.\text{sum}(0),
\end{aligned}
\tag{7}
$$

where $\boldsymbol{H} \in \mathbb{R}^{N \times d}$ is the node feature matrix and $\boldsymbol{E} \in \mathbb{R}^{N \times N \times d}$ is the edge feature matrix.

## B.2  CONDENSATION OF SELF-ATTENTION

In this section, we explain the details of the condensation of self-attention. We use $\boldsymbol{B}$ and $\hat{\boldsymbol{B}}$ to represent the self-attention matrix and its condensation.

$$
\begin{aligned}
\boldsymbol{B} &= \text{Softmax}(\frac{\boldsymbol{Q}\boldsymbol{K}^{\top}}{\sqrt{d_q}})\boldsymbol{V}, \\
\hat{\boldsymbol{B}} &= \text{Condense}(\boldsymbol{B}).
\end{aligned}
\tag{8}
$$

Since $B$ is row-normalized, we hope the condensation $\hat{B}$ is still approximately row-normalized. For this purpose, we first sum the columns of $B$ through the pre-defined frequency bands, e.g., Low, Medium, and High, and then average the rows.

$$\hat{B}_{i,j} = \sum_{\lambda_p \in f_i} \sum_{\lambda_q \in f_j} B_{p,q} \bigg/ |\mathbf{1}_{\lambda \in f_i}|, \tag{9}$$

where $f_i$ and $f_j$ are the frequency bands, and $|\mathbf{1}_{\lambda \in f_i}|$ indicates the number of eigenvalues belonging to the frequency band $f_i$.

Through this condensation strategy, we can approximately preserve the self-attention matrix's information and find the frequency bands' dependency patterns.

## C MORE EXPERIMENTAL RESULTS

### C.1 EIGENVALUE ENCODING

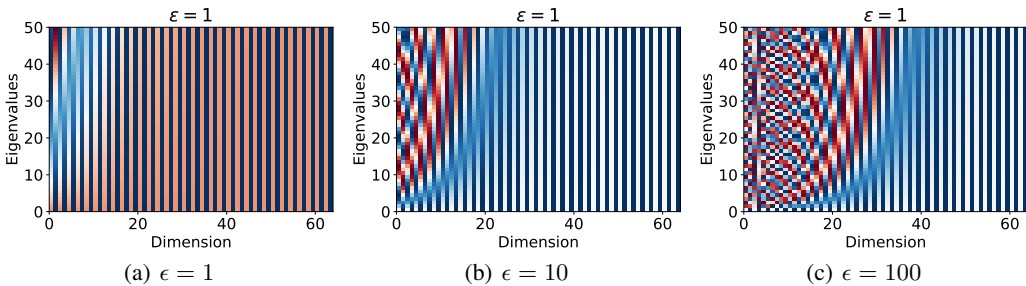

(a) $\epsilon = 1$      (b) $\epsilon = 10$      (c) $\epsilon = 100$

Figure 6: Eigenvalue encoding with different values of $\epsilon$. Best viewed in color.

Here we visualize the outputs of eigenvalue encoding with different values of $\epsilon$. Specifically, we uniformly sample 50 eigenvalues from the region $[0, 2]$ and map them into representations with $d = 64$. The results are shown in Figure 6. We can find that when $\epsilon = 1$, only the first 20 dimensions can distinguish different eigenvalues. As $\epsilon$ increases, the resolution of eigenvalue encoding becomes higher.

### C.2 TIME AND SPACE OVERHEAD

We test the time and space overheads of Specformer and two popular polynomial GNNs, *i.e.*, GPR-GNN and BernNet. Polynomial GNNs are implemented with sparse matrices and sparse matrix multiplication. We choose three datasets, *i.e.*, Squirrel, Penn94, and ZINC, two for node classification and one for graph regression. For ZINC dataset, we sample 2,000 molecular graphs as the inputs of the forward process and omit the edge features that cannot be used by polynomial GNNs.

**Setup.** In the complexity analysis, we mentioned that spectral decomposition only needs to be calculated once and can be reused in the forward process. To perform a fair comparison, we run each model for 1000 epochs and report the total time and space costs. For polynomial GNNs, we set $K = 10$ as suggested by the original papers; for Specformer, we use full eigenvectors for Squirrel and ZINC, and 6,000 eigenvectors for Penn94. The hidden dimension is $d = 64$ for all methods.

**Results.** From the time overheads in Table 9, we can find that the spectral decomposition of small graphs, *e.g.*, Squirrel and ZINC, does not bring much computational cost. And the forward time of Specformer is close to GPR-GNN and less than BernNet. This is because polynomial GNNs need to calculate $AX$ or $LX$ recurrently. Specformer only needs to calculate $U\text{diag}(\boldsymbol{\lambda})U^\top X$ once, due to the non-local capability. Besides, BernNet needs to calculate all the combinations of $L$ and $2I - L$, i.e., $\sum_{k=1}^{K} \binom{K}{k}(2I - L)^{K-k}(L)^k$, which requires a lot of computations. In the Penn94 dataset, we use truncated spectral decomposition to reduce the forward complexity from $\mathcal{O}(n^2(d + M) + nd(L + d))$ to $\mathcal{O}(q^2(d + M) + nd(L + d))$, where $q$ is the number of eigenvalues.

Table 9 shows the space overheads, where Specformer is higher than polynomials GNNs because of the dense eigenvectors. One can use fewer eigenvectors to reduce the space cost.

Table 9: Time overheads (s)

|  | GPR-GNN | BernNet | Specformer | Decomposition |
|---|---|---|---|---|
| Squirrel | 4.65 | 14.86 | 7.11 | 2.71 |
| Penn94 | 14.77 | 41.26 | 21.74 | 629.6 |
| ZINC-2k | 15.06 | 49.78 | 30.59 | 1.28 |

Table 10: Space overheads (MB)

|  | GPR-GNN | BernNet | Specformer |
|---|---|---|---|
| Squirrel | 1,257 | 1,277 | 1,845 |
| Penn94 | 3,787 | 3,799 | 4,993 |
| ZINC | 1,417 | 1,585 | 4,397 |

## C.3 SPATIAL PERSPECTIVE OF SYNTHETIC DATA

In addition to visual comparisons of learned spectral filters, we also compare Specformer and polynomial GNNs from the spatial perspective. In Figure. 7, we show the raw images, ground truth images filtered by Comb filter, *i.e.* $|\sin(\pi\lambda)|$, and images filtered by Specformer and GPR-GNN. We can see that Specformer is similar to the ground truth, and the contrast of GPR-GNN is darker than the ground truth, implying that Specformer is better than polynomial GNNs at capturing global information.

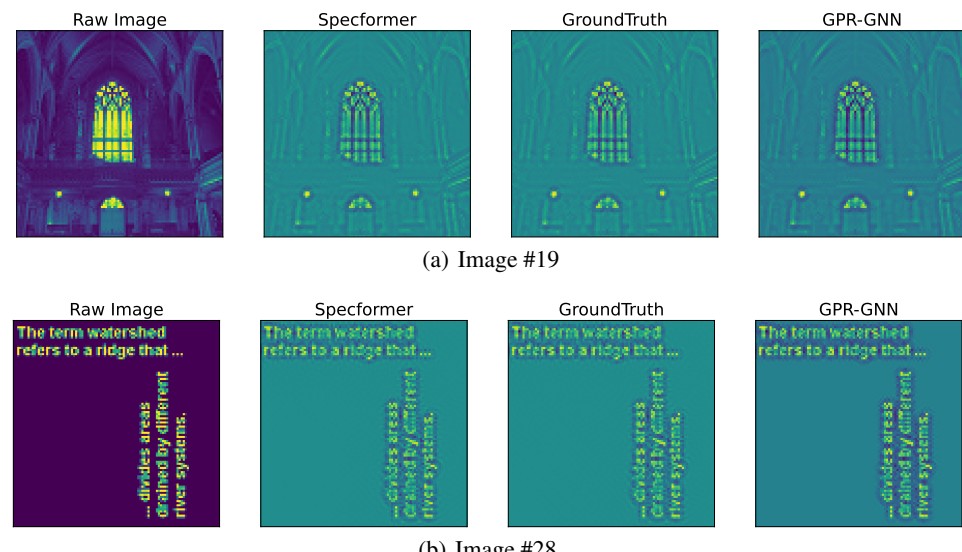

(a) Image #19

(b) Image #28

Figure 7: Synthetic data filtered by GPR-GNN and Specformer. Best viewed in color.

## C.4 EXPERIMENTS ON LARGE-SCALE MOLECULAR DATASETS.

PCQM4Mv2 is a large-scale graph regression dataset (Hu et al., 2021), which has 3.7M graphs, and the goal is to regress the HOMO-LUMO gap. We follow the experimental setting of GPS (Rampásek et al., 2022). Because the original test set is unreachable, we use the original validation set as the test set and randomly sample 150K molecules for validation

Due to the time limitation, we only run Specformer-Medium on the largest molecular datasets. The results are shown in Table 11. We can see that due to the share of learnable bases, the number of

parameters of Specformer-Medium is relatively small. That is to say, there is only one Transformer block in Specformer-Medium. But the performance of Specformer-Medium is better than the baselines with similar parameters, *e.g.*, GCN, GIN, and GPS-small.

Table 11: Results on large-scale graph dataset PCQM4Mv2.

| Model | PCQM4Mv2 | |
| --- | --- | --- |
| | MAE($\downarrow$) | Param. |
| GCN | 0.1379 | 2.0M |
| GCN-VN | 0.1153 | 4.9M |
| GIN | 0.1195 | 3.8M |
| GIN-VN | 0.1083 | 6.7M |
| GPS-small | 0.0938 | 6.2M |
| Specformer-medium | 0.0916 | 4.1M |
| GRPE | 0.0890 | 46.2M |
| EGT | 0.0869 | 89.3M |
| Graphormer | 0.0864 | 48.3M |
| GPS-medium | 0.0858 | 19.4M |

## D  THEORETICAL RESULTS

### D.1  PERMUTATION EQUIVARIANCE

**Proposition 1.** *Specformer is permutation equivariant.*

*Proof.* We show that Specformer is permutation equivariant by proving that all the components of Specformer are permutation equivariant. First, the element-wise functions, i.e., eigenvalue encoding, feed-forward networks, and layer normalization, are permutation equivariant because they are applied in node-independent manner. Second, the self-attention mechanism is permutation equivariant because $(\boldsymbol{P}\boldsymbol{Z}\boldsymbol{P}^\top)(\boldsymbol{P}\boldsymbol{Z}\boldsymbol{P}^\top)^\top = \boldsymbol{P}(\boldsymbol{Z}\boldsymbol{Z}^\top)\boldsymbol{P}^\top$, where $\boldsymbol{Z}$ is the data representation matrix and $\boldsymbol{P}$ is an arbitrary permutation matrix. Third, the construction of learnable bases is permutation equivariant because $(\boldsymbol{P}\boldsymbol{U}\boldsymbol{P}^\top)(\boldsymbol{P}\boldsymbol{\Lambda}\boldsymbol{P}^\top)(\boldsymbol{P}\boldsymbol{U}\boldsymbol{P}^\top)^\top = \boldsymbol{P}(\boldsymbol{U}\boldsymbol{\Lambda}\boldsymbol{U}^\top)\boldsymbol{P}^\top$.

Based on all the conclusions above, we prove that Specformer is permutation equivariant. $\square$

### D.2  APPROXIMATING UNIVARIATE AND MULTIVARIATE FUNCTIONS

**Theorem 1.** *(Uniform convergence of Fourier series) (Stein & Shakarchi, 2011) For any continuous real-valued function $f(x)$ on $[a, b]$ and $f'(x)$ is piece-wise continuous on $[a, b]$ and any $\epsilon > 0$, there exists a Fourier series $P(x)$ converges to $f(x)$ uniformly such that*

$$\max_{a \leq x \leq b} |P(x) - f(x)| < \epsilon. \tag{10}$$

**Theorem 2.** *(Kolmogorov–Arnold representation theorem) (Zaheer et al., 2017) Let $f : [0, 1]^M \to \mathbb{R}$ be an arbitrary multivariate continuous function. Then it has the representation*

$$f(x_1, \ldots, x_M) = \rho \left( \sum_{m=1}^{M} \lambda_m \phi(x_m) \right) \tag{11}$$

*with continuous outer and inner functions $\rho : \mathbb{R}^{2M+1} \to \mathbb{R}$ and $\phi : \mathbb{R} \to \mathbb{R}^{2M+1}$. The inner function $\phi$ is independent of the function $f$.*

**Proposition 2.** *Specformer can approximate any univariate and multivariate continuous functions.*

*Proof.* We first prove that Specformer can approximate any univariate continuous functions. We set the self-attention matrix to be an identity matrix. Then Specformer becomes a scalar-to-scalar

function, and the spectral filter is learned through the eigenvalue encoding $\rho(\lambda)$ in Equation 2. Given a linear transformation $\boldsymbol{w} \in \mathcal{R}^{d+1}$, the eigenvalue encoding becomes a Fourier series:

$$\rho(\lambda)\boldsymbol{w} = w_0\lambda + \sum_{i=1}^{d/2} w_{2i}\sin(\frac{\epsilon\lambda}{10000^{2i/d}}) + \sum_{i=1}^{d/2} w_{2i-1}\cos(\frac{\epsilon\lambda}{10000^{2i/d}}), \qquad (12)$$

where the period is determined by $\frac{\epsilon}{10000^{2i/d}}$. Because the eigenvalues fall in the range $[0, 2]$, based on Theorem 1, Specformer can approximate any univariate continuous functions in the interval $[0, 2]$. To choose orthogonal bases, one can set $\rho(\lambda, 2i) = \sin(i\lambda), \rho(\lambda, 2i+1) = \cos(i\lambda)$.

We then prove that Specformer can approximate any multivariate continuous functions. Theorem 2 states that any multivariate function is a superposition of continuous functions of a single variable. Let $\phi$ be the eigenvalue encoding, $\lambda_m$ be the self-attention weight, and $\rho$ be the FFN decoder in Equation 4. Because eigenvalue encoding can approximate any continuous univariate functions and Montanelli & Yang (2020) proves that deep ReLU networks can approximate the outer function $\rho$, Specformer can approximate any continuous multivariate functions. □

