# OpenReview forum: "Specformer: Spectral Graph Neural Networks Meet Transformers"
_ICLR.cc/2023/Conference — ICLR 2023 poster_

### Official Review · Reviewer_ur3z · 2022-10-24

**Confidence:** 4
**Correctness:** 3
**Technical Novelty And Significance:** 3
**Empirical Novelty And Significance:** 2
**Recommendation:** 6

**Clarity, Quality, Novelty And Reproducibility:**

Regarding **clarity and quality**, I have a few concerns and questions:

- C1: There are some unclear statements in the paper. For instance:
  - "spectral filters are often approximated via fixed order" , "this truncated approximation is less expressive". What do we want to approximate here?
  - "The reason is that the spectrum of Squirrel is more difficult to learn" (Section 5.4). Isn't the spectrum only the multiset of eigenvalues?

- C2: What is $\hat{S}_i$ in Eq. (6)? Does the matrix $W_x$ change with $l$? If so, I would write explicitly for precision and clarity.

- C3: How is the MLP in Eq. (5) applied to size-varying graphs in graph-level tasks? The dimensionality of $S_m$ would change with the graph size.

- C4: The paper repeatedly mentions expressivity to motivate the proposal and the adoption of Transformers. How does the expressive power of the proposed method compare to other GNNs? What are the limitations of the proposed approach?

 - C5: Does the complexity analysis (paragraph on Page 5) account for the eigendecomposition? How does the training/test time of the proposed method compare to other GNNs? Is the Laplacian eigendecomposition pre-computed? Is it negligible?

 - C6: It is hard to observe some curves (e.g., ground truth) in Figure 2.

 - C7: Experiments on graph-level tasks only consider molecular data. Is there any particular reason? The graphs in these datasets are rather small. Also, the paper does not consider very popular benchmarks for node classification from OGB. I am curious about the method's performance across graphs of very different sizes.

Regarding **originality**, to the best of my knowledge, the proposed architecture is somewhat novel.

**Reproducibility**: Code is unavailable during the reviewing process. Thus, we can not check further details and correctness. Also, some details about training (e.g., optimizer) and model selection are missing.

**Strength And Weaknesses:**


Strengths
 - Simplicity. The idea is easy-to-follow.
 - The paper introduces synthetic datasets for which it reports the approximation capability of Specformer.
 - Good performance. The proposed method yields good performance overall. For instance, it gets 0.066 MAE on the ZINC dataset.

Weaknesses
 - Limited evaluation setup. Despite the promising results, the paper lacks relevant benchmarks for node-level tasks and only considers molecular datasets (small graphs) for graph classification.
 - It is unclear whether the computational overhead of the proposed method is significant or not, and how it compares to other GNNs.
 - The paper provides limited theoretical insights about the proposed architecture.


**Summary Of The Paper:**

The paper introduces Specformer, a novel Transformer-based architecture for node- and graph-level prediction tasks. A Specformer layer embeds structure (eigenvalues) and node features separately, and then combines the obtained representations using graph convolutions. The model performs self-attention in the spectral domain. Experiments on synthetic and real-world datasets show the effectiveness of the proposed method.

**Summary Of The Review:**

Overall, the paper is relatively well-written and introduces a promising architecture for node- and graph-level prediction tasks. I have raised some concerns regarding the significance of the empirical evaluation, the computational cost/scalability, reproducibility, and clarity. Also, the paper provides limited theoretical insights about the proposed architecture, and the motivation behind some model choices is unclear/ not strong. Therefore, I am ranking this paper as marginally below the acceptance threshold.

---

> ### Author Response · Authors · 2022-11-19
> **Response to Reviewer ur3z**
>
> Thanks for your helpful comments!
>
> > **Q1: "spectral filters are often approximated via fixed order", "this truncated approximation is less expressive". What do we want to approximate here?**
>
> A1: We want to approximate the target spectral filters. For example, in the synthetic data experiment of Section 5.1, we predefined five target spectral filters, e.g., low-pass, high-pass, band-pass, band-rejection, and comb. In real-world datasets, the optimal spectral filters may not have analytic forms, but we can learn to approximate them though minimizing the loss function.
>
> >**Q2: "The reason is that the spectrum of Squirrel is more difficult to learn" (Section 5.4). Isn't the spectrum only the multiset of eigenvalues?**
>
> A2: What we mean here is that the spectral filter of Squirrel is difficult to learn. In Figure 4(d), Page 9, we visualize the spectral filter learned by Specformer. We can see that this spectral filter is in a band-rejection form and will drop quickly when the eigenvalues are close to one. Therefore, the eigenvalue encoding function is important for heterophilic graphs.
>
> >**Q3: The unclear symbols.**
>
> A3: Thanks for your reminder. At the beginning of Section 4.2, we mentioned assigning each feature dimension a separate filter can improve the performance. Therefore, $S_{i}$ is the filter assigned to the i-th feature dimension. For the symbols of $W_{x}$, we have modified it to $W_{x}^{(l-1)}$where $l$ indicates the current graph convolutional layer.
>
> >**Q4: How is the MLP in Eq. (5) applied to size-varying graphs in graph-level tasks? The dimensionality of would change with the graph size.**
>
> A4: The MLP in Eq. (5) combines the learned spectral filters, which is a channel-wise transformation. The size of MLP is related to the number of filters rather than the size of the graphs. For example, in Eq. (5), the size of filters $I, S_{1}, \cdots, S_{M}$ is $n \times n$. And we concatenate these matrices to form a tensor with size $n \times n \times (M+1)$. The MLP is performed on the last dimension. Therefore it can adopt graphs of different sizes.
>
> >**Q5: The paper repeatedly mentions expressivity to motivate the proposal and the adoption of Transformers. How does the expressive power of the proposed method compare to other GNNs? What are the limitations of the proposed approach?**
>
> A5: On Page 5, we add two propositions to show the expressiveness of Specformer. We show that Specformer can approximate any continuous functions acting on a single eigenvalue or a set of eigenvalues. That is to say, Specformer is a general and powerful univariate and multivariate function. And the polynomial GNNs are the special cases of Specformer.
>
> Our method does have some limitations. If learning optimal spectral filters requires information of full spectrum rather than top eigenvalues, then one has to perform full eigendecomposition, which could be very time-consuming on large graphs. It is better to use spatial graph convolutional approaches under this situation.
>
>
> >**Q6: Does the complexity analysis (paragraph on Page 5) account for the eigendecomposition? How does the training/test time of the proposed method compare to other GNNs? Is the Laplacian eigendecomposition pre-computed? Is it negligible?**
>
> A6: Thanks for pointing it out. In the revision, we modify the complexity analysis on Page 5. Please see the complexity and computational overheads in the general response for more information.
>
> >**Q7: It is hard to observe some curves (e.g., ground truth) in Figure 2.**
>
> A7: We have adjusted the color and shape of different curves to make the visualization readable. Please see Figure 2 in the revision.
>
> >**Q8: Experiments on graph-level tasks only consider molecular data. Is there any particular reason? The graphs in these datasets are rather small. Also, the paper does not consider very popular benchmarks for node classification from OGB. I am curious about the method's performance across graphs of very different sizes.**
>
> A8: We use molecular datasets, e.g., ZINC, MolHIV, and MolPCBA, because they are widely used benchmarks. In Table 6 of Appendix A.1, we show the smallest and largest number of nodes. Here are the results, from which we can see that MolHIV and MolPCBA datasets have graphs of different sizes.
>
> | |Average # Nodes|Minimum # Nodes|Maximum # Nodes|
> |-|-|-|-|
> |MolHIV|25.5|2|222|
> |MolPCBA|26.0|1|332|
>
> We added experiments on two large datasets. Please see the scalability part of our general response.
>
>
> >**Q9: Reproducibility**
>
> A9: In Appendix A.2, we report the experimental setting in detail, including data splitting, optimizer, model selection, environment, and hyperparameters. Besides, **we have updated our code in the supplemental material.**

---

### Official Review · Reviewer_JCjn · 2022-10-25

**Confidence:** 5
**Correctness:** 4
**Technical Novelty And Significance:** 4
**Empirical Novelty And Significance:** 4
**Recommendation:** 6

**Clarity, Quality, Novelty And Reproducibility:**

This paper is easy to follow and the clarity is good. Overall, the quality is good for an ICLR paper. The novelty is pretty good. No code is provided to evaluate reproducibility.

**Strength And Weaknesses:**

Strength
1. The idea is novel because all of spectral GNNs only focus on the spectral filter on the single eigenvalues. And it is very easy to find the rationality, even if the whitening is used in images, we still need nxnxk convolution filter rather than nxnx1 convolution filter. Thus I believe the motivation is reasonable, and we all ignore this.
2. The proposed method also inspires me. Although it still has some drawbacks but still surprises me.
3. Comprehensive experiments are convincing in my opinion.

Weaknesses:
1. I am curious if the set2set filter is able to find some interesting patterns from a spatial perspective. Because existing experiments are so quantitive, is there any possibility of digging up some quantitative experiments to demonstrate what kind of pattern the proposed method is able to find and the scalar-based spectral filter cannot?
2. The computational complexity is quite high because of eigenvalues. So is it possible to explicitly use eigenvalues rather than implictly work on eigenvalues? Because I notice the experiments on small datasets. Although this method is promising, I still hope there is any possibility for a large-scale version.

**Summary Of The Paper:**

In this paper, authors challenge a usual way of designing a spectral function with a dimensional-wise mapping on Graph diffusion. Based on this, authors pay more attention on the global pattern of the specturm rather than single eigenvalue and propose a learnable set2set spectral filter. Based on the new spectral filter, more powerful spectral graph neural network is built. In experiments, authors demonstrate the convincing result for the proposed method.

**Summary Of The Review:**

This paper is very interesting, although there are some shortcomings in implementations. It opens an promising direction for GNN. I am optimistic about these issues. Thus I make the recommendation to accept.

---

> ### Author Response · Authors · 2022-11-19
> **Response to Reviewer JCjn**
>
> Thanks for your positive comments!
>
> >**Q1: I am curious if the set2set filter is able to find some interesting patterns from a spatial perspective. Because existing experiments are so quantitive, is there any possibility of digging up some quantitative experiments to demonstrate what kind of pattern the proposed method is able to find and the scalar-based spectral filter cannot?**
>
> A1: Explaining the results from the spatial perspective is a good point and could help people understand the advantages intuitively. In Appendix C.3, we add an experiment to show the synthetic data filtered by set2set (Specformer) and scalar2scalar (GPR-GNN). Because the synthetic data are generated from images, we visualize the images filtered by the two methods and the ground truth. We find that the background of images filtered by GPR-GNN is darker than the ground truth (having lower contrast), and images filtered by Specformer are close to the ground truth. This discovery may suggest that Specformer is better than polynomial GNNs at capturing global information.
>
> >**Q2: The computational complexity is quite high because of eigenvalues. So is it possible to explicitly use eigenvalues rather than implicitly work on eigenvalues?**
>
> A2: Specformer does explicitly take eigenvalues as input. To reduce the computational cost, we can use truncated spectral decomposition to calculate the top eigenvalues, as discussed in the general response and Section 5.2 of the revision.
>
> >**Q3: Hope there is any possibility for a large-scale version.**
>
> A3: Please see the scalability part of our general response.

---

### Official Review · Reviewer_T8vi · 2022-10-26

**Confidence:** 4
**Clarity, Quality, Novelty And Reproducibility:** 1. The author introduces some weaknes…
**Correctness:** 2
**Technical Novelty And Significance:** 2
**Empirical Novelty And Significance:** 2
**Recommendation:** 5

**Strength And Weaknesses:**

1. The author introduces some weaknesses for current spectral fitler, but the explaination is not convincing enough.
1) At first, the authors propose -- "mapping a single eigenvalue to a single filtered value, thus
ignoring the global pattern of the spectrum", which means that the interaction among eigenvalues are lacked. However, how to understand the interaction among different eigenvalues (frequiences in Graph Signal Processing)?
I think that this assumption should be supported via theoretical analysis.
2) Secondly, the authors propose -- "these filters are often constructed based on some fixed-order polynomials, which have limited expressiveness and flexibility".
In fact, the polynomials paradigm was proposed to avoid the huge cost of eigenvalue decomposition for calculating U of Laplacian matrix.
However, the proposed model needs to calculate U, which introduces huge cost. In other words, we can also learn the fiter parameters based on other networks (without transformer) if we ignore the cost of eigenvalue decomposition.
Thus I think the proposed model does not handle the trade-off between computation cost and expressiveness.
2. I also have some questions for the architecture of proposed model, includes,
1) How to understand the benifits of Eq.(2)-the EE form which encodes the eigenvalues?
2) The proposed model needs to calculate all eigenvectors U and eigenvalues (O(n^3)), the complexity analysis is not true.

**Summary Of The Paper:**

This paper proposes a spectral graph filter via introducing transformer structure, the idea seems novel, but the following points need to be concerned and explained.


**Summary Of The Review:**

This paper proposes a spectral graph filter via introducing transformer structure, the idea seems novel, but the following points need to be concerned and explained. The author introduces some weaknesses for current spectral fitler, but the explaination is not convincing enough. How to understand the benifits of Eq.(2)-the EE form which encodes the eigenvalues? The proposed model needs to calculate all eigenvectors U and eigenvalues (O(n^3)), the complexity analysis is not true. More questions can been seen above.

---

> ### Author Response · Authors · 2022-11-19
> **Response to Reviewer T8vi**
>
> Thanks for your helpful comments!
>
> >**Q1: How to understand the interaction among different eigenvalues (frequiences in Graph Signal Processing)? I think that this assumption should be supported via theoretical analysis.**
>
> A1: In the revision, we add two propositions on Page 5.
>
> Proposition 2 shows that Specformer can approximate any multivariate functions acting on a set of eigenvalues. We prove this by showing that with the help of self-attention (interaction among different eigenvalues), Specformer meets the form of $\rho\left(\sum_{m=1}^M \lambda_m \phi\left(x_m\right)\right)$, where $\lambda_m$ is the attention weights. And Specformer is a general and powerful multivariate function. This multivariate function could possibly learn some important information from a set of eigenvalues, e.g., the algebraic multiplicity of the eigenvalue 0 and spectral gap.
>
> >**Q2: Secondly, the authors propose -- "these filters are often constructed based on some fixed-order polynomials, which have limited expressiveness and flexibility". In fact, the polynomials paradigm was proposed to avoid the huge cost of eigenvalue decomposition for calculating U of Laplacian matrix. However, the proposed model needs to calculate U, which introduces huge cost.**
>
> A2: The motivation of using polynomials is indeed to bypass the eigendecomposition. However, this is only an issue with full decomposition on large graphs.
>
> 1. We conduct experiments on the time and space overheads to compare Specformer with polynomial GNNs, e.g., GPR-GNN and BernNet. Results are shown below. We can see that on small graphs, the decomposition does not bring significant computational costs. On the contrary, BernNet costs more time than Specformer. In addition, the spectral decomposition only needs to be calculated once and can be used repeatedly. Therefore, if we train the model for many iterations, the forward-pass cost is more than the decomposition.
>
> Time overheads (s)
> | | GPR-GNN | BernNet | Specformer | Decomposition |
> | - | - | - | - | - |
> | Squirrel | **4.65** | 14.86 | 7.11 | 2.71 |
> | Penn94 | **14.77** | 41.26 | 21.74 | 629.61 |
> | ZINC-2k | **15.06** | 49.78 | 30.59 | 1.28 |
>
> 2. For large graphs, we do not necessarily need all eigenvalues. Instead, we can exploit fast numerical algorithmis like Krylov subspace methods to estimate the top K eigenvalues. We also add experiments on large graphs to support this argument. Please see the scalability part of our general response.
>
> >**Q3: In other words, we can also learn the fiter parameters based on other networks (without transformer) if we ignore the cost of eigenvalue decomposition. Thus I think the proposed model does not handle the trade-off between computation cost and expressiveness.**
>
> A3: Other neural networks could be used. But we choose Transformer as the spectral filter because it is a set-to-set function, permutation equivariant, and inductive. These good properties cannot be simply replaced by other networks. For example, we can use an MLP as the spectral filter, but it is sensitive to the node permutation and can only be used for the transductive setting.
>
> >**Q4: How to understand the benifits of Eq.(2)-the EE form which encodes the eigenvalues?**
>
> A4: At the bottom of Page 3 and top of Page 4, we introduce the benefits of eigenvalue encoding. In short, eigenvalue encoding provides vector representations of eigenvalues (scalars) which (1) facilitate learning of the relative frequency shifts of eigenvalues and (2) provide a multi-scale representation. Besides, on Page 5, Proposition 1, we prove that the eigenvalue encoding is essentially a Fourier series, which can approximate any functions acting on the eigenvalues.
>
> >**Q5: The complexity analysis is not true because of the complexity of decomposition.**
>
> A5: We have modified our complexity analysis on Page 5 and explained it in our general response. Specformer has two parts of calculation: decomposition and forward process. Spectral decomposition is pre-computed and has the complexity of $\mathcal{O}(n^3)$. The overall forward complexity is $\mathcal{O}(n^{2}(d+M) + nd(L+d))$.

---

### Official Review · Reviewer_Ge61 · 2022-10-26

**Confidence:** 4
**Correctness:** 3
**Technical Novelty And Significance:** 3
**Empirical Novelty And Significance:** 2
**Recommendation:** 5

**Clarity, Quality, Novelty And Reproducibility:**

Clarity: The paper is overall reader-friendly.

Quality and Novelty: This paper has a clear motivation, and the proposed method is marginally novel.

Reproducibility: There is no available code.


**Strength And Weaknesses:**

Strengths：

1)	The motivation for using self-attention in the spectral domain is clear and inspiring.

2)	The authors' theoretical comparisons of Specformer to Polynomial GNNs, MPNNs, and Graph Transformers indicate Specformer's flexibility and universality, which looks sound to me.

3)	Experiments demonstrate that Specformer can learn a flexible and expressive filter and outperform baselines on node-level and graph-level tasks, particularly on heterophilic graph datasets.

4)	This paper is well-written and easy to follow.

Weaknesses:

1)	The scalability of Specformer is seriously limited by its quadratic time complexity.

2)	It would be helpful to empirically compare Specformer with baselines in terms of time and space overhead, even as the Specformer's time complexity is theoretically analyzed.

3)	Some baseline experimental results are missing, and no explanation. For example, the results of the Graphormer on MolHIV and MolPCBA are not displayed in Table 3 but are reported in the Graphormer's paper.

4)	The experimental datasets are small and a bit old. Using large and latest datasets (such as ogbn-arxiv in OGB and Penn94 in LINKX [1]) may significantly enhance the paper's quality.

[1] Lim, Derek, et al. "Large scale learning on non-homophilous graphs: New benchmarks and strong simple methods." In NeurIPS 2021.


**Summary Of The Paper:**

This paper proposes Specformer, a Transformer-based graph spectral filter that captures the magnitudes and relative dependencies of all Laplacian eigenvalues. Specformer is permutation equivariant and can perform non-local graph convolutions. Extensive experiments on the node-level and graph-level datasets demonstrate Specformer's promising performance.

**Summary Of The Review:**

The motivation of this paper is clear, and the proposed method is promising. However, Specformer's quadratic time complexity is a serious limitation, and the experimental results are incomplete since some baseline results are missing. Due to the negative aspects of this work, I am slightly inclined to recommend rejection, but I would encourage the authors to address some of the issues raised in the comments in the rebuttal.

---

> ### Author Response · Authors · 2022-11-19
> **Response To Reviewer Ge61**
>
> Thank you for the helpful comments!
>
> > **Q1: The scalability of Specformer is seriously limited by its quadratic time complexity.**
>
> A1: The quadratic time complexity is caused by the self-attention module. To improve the scalability, we use truncated spectral decomposition to calculate the top-$q$ important eigenvectors, and the complexity will reduce from $\mathcal{O}(n^2)$ to $\mathcal{O}(q^2)$. In the future, we will try to further reduce the complexity by sparsifying the self-attention matrix, e.g., deformable attention.
>
> > **Q2: It would be helpful to empirically compare Specformer with baselines in terms of time and space overhead.**
>
> A2: Please see our general response on the time and space overheads. We also include more details in Appendix C.2 of the revision.
>
> >**Q3: Some baseline experimental results are missing, and no explanation, e.g., Graphormer**
>
> A3: Graphormer is first pre-trained on a large dataset PCQM4M-LSC [1] and then fine-tuned on MolPCBA and MolHIV with an additional graph augmentation trick FLAG [2] to prevent the model from over-fitting. However, existing methods do not follow this setting but directly train the models from the sketch. For example, the most recent work, GPS [3]. Therefore, we did not report the performance of Graphormer in the original version. As for other baselines, if the original papers do not perform experiments on ZINC, MolHIV, and MolPCBA datasets, we do not report their performances.
>
> [1] https://ogb.stanford.edu/neurips2022/
>
> [2] Flag: Adversarial data augmentation for graph neural networks.
>
> [3] Recipe for a General, Powerful, Scalable Graph Transformer. NeurIPS 2022.
>
> **In the revision, we re-ran Graphormer without pre-training and augmentation and updated the results of Table 3.** Here are the results. $\downarrow$ means lower the better and $\uparrow$ means higher the better.
>
> | | ZINC ($\downarrow$) | MolHIV ($\uparrow$) | MolPCBA ($\uparrow$) |
> | - | - | - | - |
> | Graphormer | 0.122 ± 0.006 | 0.7640 ± 0.0022 | 0.2643 ± 0.0017 |
> | Specformer | **0.066 ± 0.003** | **0.7889 ± 0.0124** | **0.2972 ± 0.0023** |
>
> From the table, we can see that our Specformer outperforms the Graphormer on three datasets.
>
> >**Q4: The experimental datasets are small and a bit old.**
>
> A4: Please see the scalability part of the general response.

---

### Public Comment · ~Benedek_Andras_Rozemberczki1 · 2022-11-05
**Misattribution of certain datasets**

The paper misattributed the authorship of the Chameleons and Squirrels datasets. These datasets were proposed in this ICLR submission:

https://openreview.net/forum?id=HJxiMAVtPH

The Pei et al. paper cited by the authors took the Squirrel and Chameleons datasets and used those for benchmarking, but had nothing to do with the creation of the datasets. The correct citation for the paper which proposed the datasets is:

```bibtex
>@article{musae,
          author = {Rozemberczki, Benedek and Allen, Carl and Sarkar, Rik},
          title = {{Multi-Scale Attributed Node Embedding}},
          journal = {Journal of Complex Networks},
          volume = {9},
          number = {2},
          year = {2021},
}
```

---

> ### Author Response · Authors · 2022-11-19
> **Response to the citation of datasets**
>
> Thanks for pointing out the issue of citation.
>
> We fix it and cite your paper. See Section 5.2 for details.

---

### Author Response · Authors · 2022-11-19
**General Response From Authors**

We thank all reviewers for their valuable feedback and suggestions. We have addressed most of the questions and suggestions in the revised paper (highlighted in blue color). We respond to the common concerns here and reply to individual reviewers for other questions.

> **Complexity**

We thank Reviewer T8vi and ur3z for pointing out the complexity of spectral decomposition. We revise the complexity analysis on Page 5.

The spectral decomposition is pre-computed and has the complexity of $\mathcal{O}(n^3)$ and the complexity of the forward pass of our model is $\mathcal{O}(n^{2}(d+M) + nd(L+d))$. Since we only need to perform decomposition once per graph, the overall complexity of Specformer is the sum of the forward complexity and the decomposition complexity amortized over the number of uses in training and inference, rather than a simple summation of the two.

> **Scalability**

We thank Reviewer Ge61, JCjn, and ur3z for suggesting us to test our model in large graphs to validate the scalability of Specformer.
We use the recommended two large node classification datasets, e.g., Penn94 (41,554 nodes) [1] and ogbn-arXiv (169,343 nodes) [2], for heterophilic and homophilic settings. Here we only report part of the results. The complete results can be found in Table 2 in the revision.

Specifically, to apply Specformer to large graphs, we use truncated spectral decomposition, which only calculates $q$ eigenvectors, and the complexity reduces to $\mathcal{O}(q^{2}(d+M) + nd(L+d))$.
Based on the filters learned in the small datasets, we find that band-rejection filters are important for heterophilic datasets and low-pass filters are suitable for homophilic datasets. See Figures 4(b) and 4(d) for details. Therefore, we use the eigenvectors with the smallest 3000 (low-frequency) and largest 3000 (high-frequency) eigenvalues for Penn94 and eigenvectors with the smallest 5000 eigenvalues (low-frequency) for ogbn-arXiv.

| | GCN | GCNII | GPR-GNN | BernNet | ChebNetII | Specformer |
| - | - | - | - | - | - | - |
| Penn94 | 82.47±0.27 | 82.92±0.59 | 81.38±0.16 | 82.47±0.21 | 83.12±0.22 |**84.32±0.32** |
| arXiv | 71.74±0.29 | 72.04±0.19 | 71.78±0.18 | 71.96±0.27 | 72.32±0.23 | **72.37±0.18** |

Moreover, we ran an experiment on the PCQM4Mv2 [3] dataset, which contains 3,746,620 molecular graphs.

| | GCN | GCN-VN | GIN | GIN-VN | GPS-small |Specformer-Medium|
|-|-|-|-|-|-|-|
|PCQM4Mv2|0.1379|0.1153|0.1153|0.1083|0.0938|**0.0917**|
|Param.|2.0M|4.9M|3.8M|3.7M|6.2M|4.1M|

[1] Large scale learning on non-homophilous graphs: New benchmarks and strong simple methods. NeurIPS 2021.

[2] Open graph benchmark: Datasets for machine learning on graphs. NeurIPS 2020.

[3]  OGB-LSC: A large-scale challenge for machine learning on graphs. NeurIPS 2021.

> **Computational Overheads**

We thank reviewers Ge61 and ur3z for suggesting adding experiments on computational overheads. We test the time and space overheads of Specformer and two popular polynomial GNNs, i.e., GPRGNN and BernNet, in three datasets, i.e., Squirrel, Penn94, and ZINC. See Appendix C.2 for details.

Because spectral decomposition only needs to be calculated once and can be reused in training, to perform a fair comparison, we run each model 1000 epochs and report the total time and space costs. Polynomial GNNs are implemented with sparse matrices and sparse matrix multiplication. For ZINC dataset, we sample 2,000 molecular graphs as the inputs. We use full eigenvectors for Squirrel and ZINC, and 6,000 eigenvectors for Penn94. The hidden dimension is d = 64 for all methods.

Time overheads (s)
| | GPR-GNN | BernNet | Specformer | Decomposition |
| - | - | - | - | - |
| Squirrel | **4.65** | 14.86 | 7.11 | 2.71 |
| Penn94 | **14.77** | 41.26 | 21.74 | 629.61 |
| ZINC-2k | **15.06** | 49.78 | 30.59 | 1.28 |

Space overheads (MB)
| |GPR-GNN|BernNet|Specformer|
|-|-|-|-|
|Squirrel|**1,257**|1,277|1,845|
|Penn94|**3,787**|3,799|4,993|
|ZINC-2k|**1,417**|1,585|4,397|

We can see that the spectral decomposition of small graphs, e.g., Squirrel and ZINC, does not bring much computational cost. And the forward time of Specformer is close to GPR-GNN and less than BernNet. This is because polynomial GNNs need to calculate $AX$ or $LX$ recurrently. Specformer only needs to calculate $U \text{diag}(\lambda) U^{\top}$ once. BernNet needs to calculate all the combinations of $L$ and $2I-L$, i.e., $\sum_{k=1}^{K} \tbinom{K}{k}(2I-L)^{K-k}(L)^{k}$, which requires a lot of computations. Besides, In the Penn94 dataset, we use truncated spectral decomposition to reduce the forward complexity to $\mathcal{O}(q^{2}(d+M) + nd(L+d))$, where $q$ is the number of eigenvectors.

> **Reproducibility**

**We include our code in the supplemental material.**

---

### Public Comment · ~Anson_Bastos1 · 2023-02-15
**Misattribution of a graph transformer phenomenon in the paper**

Firstly we thank and congratulate the authors for presenting a novel way of learning functions in graph spectral space using transformers.

We would like to bring the authors' attention to a point in the paper that is not adequately cited. The paper mentions that graph transformers have a low pass phenomenon and attributes this result to Wang et al, Shi et al. These works however only show results for the computer vision and natural language domain and not for general graphs. This phenomenon for general graph structured data is shown in the below paper:

https://openreview.net/forum?id=aRsLetumx1

Shi et al analyses the response of the layer norm module and compares it with a rank one matrix (for language tokens). Similarly Wang et al analyse the frequency components for the regular grid structured data (not for the generic graph spectra). In addition to the these works (which address special cases), the appropriate citation for the paper which shows graph transformer has low pass characteristics would be:

```
@article{
	bastos2022FeTA,
	title={How Expressive are Transformers in Spectral Domain for Graphs?},
	author={Anson Bastos and Abhishek Nadgeri and Kuldeep Singh and Hiroki Kanezashi and Toyotaro Suzumura and Isaiah Onando Mulang'},
	journal={Transactions on Machine Learning Research},
	year={2022},
	url={https://openreview.net/forum?id=aRsLetumx1},
	note={}
}
```

We humbly request the authors to faithfully cite this work in addition to the ones already cited.

Thanks

---

> ### Author Response · Authors · 2023-02-16
> **Thanks for the reference**
>
> Thanks for pointing out the relevant paper. We will take a look and discuss it in our final version.

---

### Decision · Program_Chairs · 2023-01-20

**Decision:**

Accept: poster

**Justification For Why Not Higher Score:**

This paper presents some very impressive results. However, the results are not fully reproducible with the submitted code.

**Justification For Why Not Lower Score:**

N/A

**Metareview: Summary, Strengths And Weaknesses:**

This paper proposes to use a set to set transformer architecture over eigenvalues of graph Laplacian to learn more expressive spectral filters. Compared to the polynomial parameterization used in classical spectral GNNs such as Chebynet, transformer over eigenvalue set can model their dependencies and can approximate wider filter functions, especically considering that polynomial functions cannot handle multiple eigenvalues and must give them the same filtering (even they represent different frequencies). It is a pity that the authors did not point it out in their analysis and motivation. Nevertheless, the expressivity is at the cost of doing eigenvalue decomposition, which is impossible for large networks. So a cutoff of largest and smallest K eigenvalues are used for large networks.

I want the authors know that this is identified as a borderline paper and a reviewer-AC meeting is conducted to discuss the paper. Through the discussion, both reviewers and I agree that the motivation of using transformer makes much sense (though the authors did not fully reveal where the additional expressivity comes from compared to polynomial GNNs) and the experimental results are impressive (both ZINC and Squirrel are new SoTA). To verify the results, one reviewer and I volunteered to reproduce the results using the submitted code. The reviewer reported that with some effort, they successfully reproduced the ZINC results with 0.0678 MAE in 500 epochs (the original takes 1000 epochs). However, on my side, the node-level code provided by the authors cannot run. There is no README, no preprocessed data, no script for preprocessing datasets other than arxiv and Penn, and no comments or direction in the jupyter notebook. I believe for papers with such significant results, a reproducibility check is a must. However, the submitted code seems not in a fully runnable state.

Considering all above, I still believe that the strengths outweigh the weaknesses. This paper can inspire new ways to design spectral GNNs and has impressive empirical results. So I recommend an accept but strongly encourage the authors to polish and complement the code and ensure the reproducibility of the experiments.

**Note From Pc:**

if the above contains the word "oral" or "spotlight" please see: "oral" presentation means -> notable-top-5% and "spotlight" means -> notable-top-25%. As stated in our emails, we are disassociating presentation type from AC recommendations

**Summary Of Ac-Reviewer Meeting:**

Through the discussion, both reviewers and I agree that the motivation of using transformer makes much sense (though the authors did not fully convey it) and the experimental results are impressive (both ZINC and Squirrel are new SoTA). To verify the results, one reviewer and I volunteered to reproduce the results using the submitted code. However, from my side, the node-level code provided by the authors cannot run. There is no README, no preprocessed data, no script for preprocessing datasets other than arxiv and Penn, and no comments or direction in the jupyter notebook. I believe for papers with such significant results, a reproducibility check is a must. However, the submitted code seems only to satisfy the code requirement, but is not in a fully runnable state.